# Preconception syphilis seroprevalence and association with duration of marriage and age among married individuals in Guangdong Province, China: A population-based cross-sectional study

**Wenxue Xiong**[1☉], **Lu Han**[2☉], **Rui Li**[1☉], **Xijia Tang**[1], **Chaonan Fan**[1], **Xiaohua Liu**[2], **Jiabao Wu**[2], **Hua Nie**[2], **Weibing Qin**[2]*, **Li Ling**[1,3]*

**1** Faculty of Medical Statistics, School of Public Health, Sun Yat-sen University, Guangzhou, Guangdong, China, **2** NHC Key Laboratory of Male Reproduction and Genetics, Guangdong Provincial Reproductive Science Institute (Guangdong Provincial Fertility Hospital), Guangzhou, Guangdong, China, **3** Clinical research design division, Clinical research center, Sun Yat-sen Memorial Hospital, Sun Yat-sen University, Guangzhou, Guangdong, China

☉ These authors contributed equally to this work.
* guardqin@163.com (WQ); lingli@mail.sysu.edu.cn (LL)

**Data Availability Statement:** The data analyzed in our study are not publicly available because of the

## Abstract

### Background

Duration of marriage (DoM) and age are important characteristics of married individuals, who are the critical population for eliminating mother-to-child transmission (MTCT) of syphilis. A deep understanding of the preconception syphilis seroprevalence (PSS) and its distribution among this population may be able to help to eliminate MTCT. However, few population-based epidemiological studies have been focused on this group, and the association of DoM and age with PSS remains unclear.

### Methodology/Principal findings

This study used data from 4,826,214 married individuals aged 21–49 years who participated in the National Free Preconception Health Examination Project in Guangdong Province, China, between 2014 and 2019. Syphilis was screened using the rapid plasma reagin (RPR) test. The seroprevalence time series, seroprevalence map, and hot spot analysis (HSA) were employed to visualize the spatiotemporal distribution. The restricted cubic spline (RCS) based on multivariate logistic regression was used to model the association of DoM and age with PSS. The interactions on the additive scale of DoM and age were also assessed.

The PSS was 266.61 per 100,000 persons (95% CI: 262.03–271.24) and the burden was higher in economically underdeveloped area within the province. A strong J-shaped non-linearity association was observed between age and PSS. Specifically, the risk of seropositivity was relatively flat until 27 years of age among men and increased rapidly afterwards, with

limitation of data availability in the data management rule of Guangdong Provincial Fertility Hospital. Access to these data may be requested through the National Health Commission Key Laboratory of Male Reproduction and Genetics, China (contact via 87651111@163.com) for researchers who meet the criteria for access to confidential data.

**Funding:** The research was funded by the Guangdong Province Medical Research Funding (No.2022314) to LH, the Guangdong Province Medical Research Funding (No.2022326) to XL, and the Natural Science Foundation of Guangdong Province (No. 2019A1515011984) to WQ. The funders had no role in study design, data collection and analysis, decision to publish, or preparation of the manuscript.

**Competing interests:** The authors have declared that no competing interests exist.

an adjusted odds ratio (aOR) of 1.13 (95% CI: 1.12–1.13) per unit. Among women, the risk of seropositivity was relatively flat until 25 years of age and increased rapidly afterwards with an aOR of 1.08 (95% CI: 1.08–1.09) per unit. DoM was negatively associated with PSS among married individuals. Moreover, the combined effects of age and DoM appeared to be synergistic.

## Conclusions/Significance

Our findings suggest that attention should be paid to preventing syphilis in underdeveloped areas and that syphilis screening in newly married individuals who are in their late 20s or older should be recommended. Additionally, early syphilis prevention strategies should be implemented among young people as early as possible.

## Author summary

Preconception screening for syphilis in nonpregnant populations is an important public health approach to prevent the sexual transmission and subsequent mother-to-child transmission (MTCT) of syphilis. Duration of marriage (DoM) and age are important characteristics of married individuals, who are the critical population for eliminating MTCT. A deep understanding of the preconception syphilis seroprevalence (PSS) and its distribution among this population may be able to help to eliminate MTCT. Using data from about 4.8 million married individuals aged 21–49 years in Guangdong Province, we found that the burden of syphilis was relatively low among married individuals in Guangdong Province, and more attention should be paid to underdeveloped areas. There may be a J-shaped relationship between age and PSS, suggesting that after a certain age, PSS increases rapidly with age. DoM was negatively associated with PSS, and older newly married women were more likely to be seropositive. These results suggest that educating people in early adulthood about syphilis prevention may mitigate the increasing cost of syphilis to the health system. Screening for syphilis in newly married individuals who are in their late 20s or older should be recommended. Medical professionals could advise pregnant women's male spouses to undergo syphilis testing as well.

## Introduction

Syphilis is a type of disease caused by *Treponema pallidum* that causes a great public health burden. According to World Health Organization, the global prevalence of maternal syphilis in 2016 was estimated at 0.69%, resulting in 661,000 total cases of congenital syphilis (CS) and 355,000 associated adverse pregnancy outcomes [1]. Syphilis has been endemic again in China since the 1980s [2], and CS has reemerged as a common, preventable cause of adverse pregnancy outcomes [3], with previous studies showing the highest incidence in people of reproductive age (21–49 years) [4–6]. However, the maternal prenatal screening seems inadequate for eliminating mother-to-child transmission (MTCT) [7]. A national study showed that in rural China, the preconception syphilis seroprevalence (PSS) among women was 0.37% [8], but the incidence of syphilis was 0.025% [9]. The pre-exiting syphilis infection accounts for a large proportion of syphilis infection. To further eliminate MTCT and reduce the risk of adverse pregnancy outcomes, preconception syphilis screening is considered an option and is

recommended by both the US Preventive Services Task Force and the US Center for Disease Control and Prevention (CDC) [10,11]. When deciding who to screen for syphilis, clinicians should know the prevalence of syphilis infection in the communities they serve and other sociodemographic factors that may be associated with an increased risk of syphilis infection [11].

Although syphilis is curable [12], once an individual was infected, treponemal antibodies will remain in the human body throughout life [13]. The rapid plasma reagin (RPR) test is therefore used to assess the burden of pre-existing syphilis infection [7,14]. Several studies suggested that age was closely related to syphilis seroprevalence [8,10]. However, these studies all treated age as a categorical variable, which could lead to information loss, power reduction, and complexities in analysis [15]. Instead of categorizing continuous variables, restricted cubic spline (RCS) functions can characterize a dose-response relationship between a continuous exposure and an outcome [16]. Duration of marriage (DoM) is also an important characteristic of married individuals [17], but few studies have examined the association between DoM and PSS [18], let alone its interaction with age. Exploring the characteristics associated with PSS can help identify high-risk groups, provide recommendations for early syphilis prevention, and further facilitate the development of effective responses [19]. Therefore, the association of DoM and age with PSS among married individuals of reproductive age needs to be further explored.

Quantifying regional PSS can help identify high-risk areas, allowing for a more efficient allocation of health services and resources [20]. However, researchers tend to focus more on vulnerable groups, such as female sex workers and men who have sex with men [21,22]. A limited number of studies have expounded on the epidemiological features of syphilis among married individuals of reproductive age [8]. But most of these studies are based on mandatory reporting systems rather than population-based [23], which is considered the gold standard in evaluating disease epidemics [7].

Thus, this study aimed to explore the PSS and spatiotemporal distribution among married individuals aged 21–49 years in Guangdong Province, China and to investigate the association of DoM and age with PSS.

## Method

### Ethics statement

Each participant provided written informed consent prior to enrollment and the project was approved by the Institutional Review Board of the Chinese Association of Maternal and Child Health Studies (IRB-201001).

### Study area

Guangdong Province, located in South China (Fig 1), witnessed a rapid increase in reported syphilis cases from 55,936 in 2014 to 62,760 in 2019 [24]. Approximately 22.4 million married women of reproductive age live in Guangdong Province [24]. According to geographic location and economic development, Guangdong was divided into four regions: the Pearl River Delta, East Wing, West Wing, and Mountainous Area (Fig 1) [25]. There was a high degree of heterogeneity in internal economic conditions, demographic composition, and meteorological conditions between the four regions of Guangdong province, a microcosm of China [23,25]. The National Free Preconception Health Examination Project (NFPHEP) provided preconception counseling, education, and medical examinations to legally married couples planning a pregnancy within the next six months, which included free syphilis screening for both

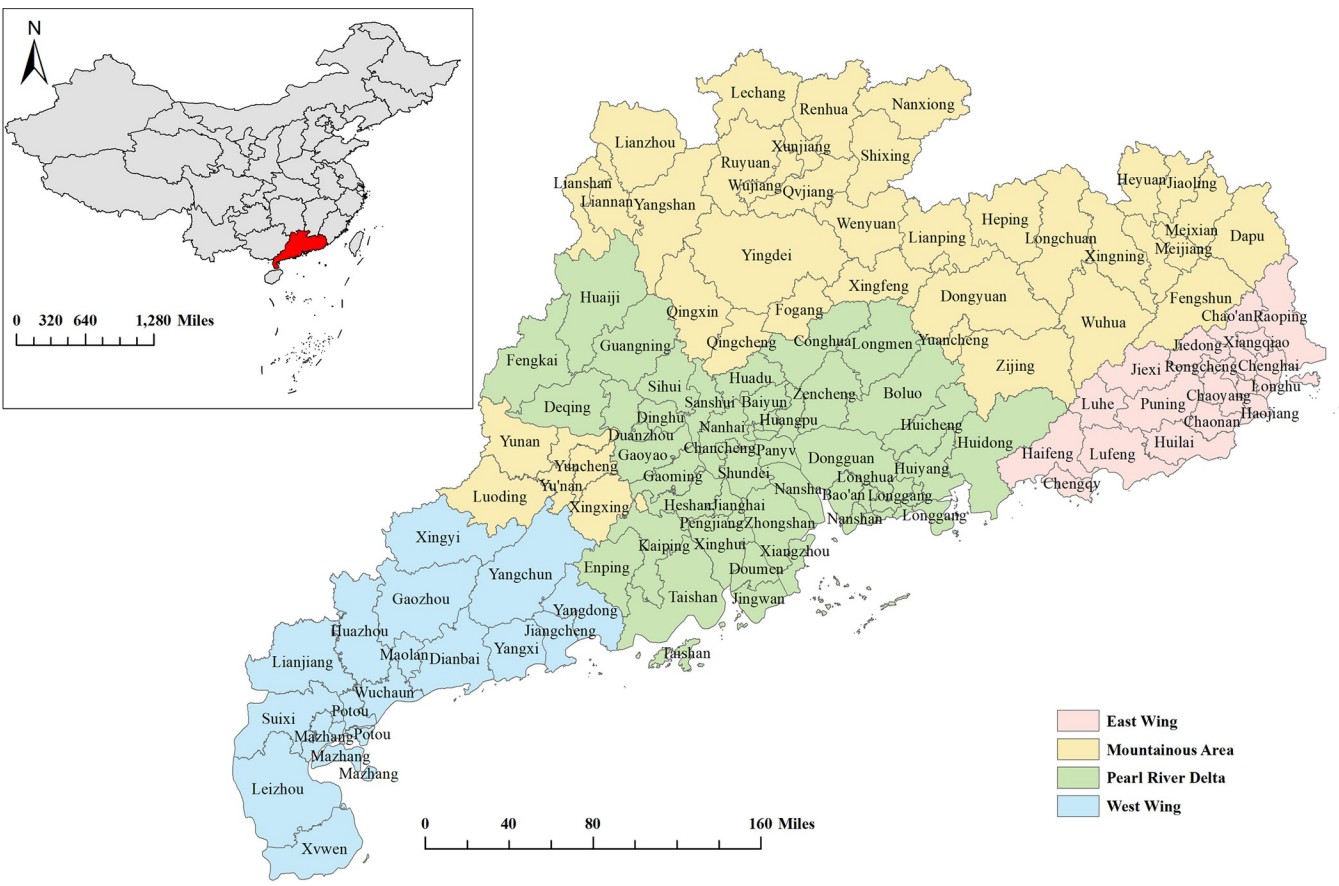

**Fig 1. The location of Guangdong Province and the division of its four regions.** Base layer of the map was downloaded from Resource and Environment Science and Data Center (https://www.resdc.cn/data.aspx?DATAID=201).

spouses. And this project was well implemented in Guangdong Province, with a coverage rate of over 85% of the target population [24], providing sufficient data for this study.

## Study design and participants

**Study design.** This was a population-based cross-sectional study that used data of married individuals enrolled in the NFPHEP in Guangdong Province, China, from January 2014 to November 2019. The National Health and Family Planning Commission of the People's Republic of China launched the NFPHEP nationwide in 2010. Couples planning a pregnancy are volunteering to participate in the NFPHEP. In order to recruit targeted participants, the government set up counseling stations at marriage registration offices, held lectures in the communities/districts, and used various forms of media, such as television and newspapers, to reach these couples. The NFPHEP has also been implemented by providing special staff to inform couples of the project when they apply for the Family Planning Service Certificate, which is a must if couples plan to have children in China [26]. Then those couples who come forward to consult on the NFPHEP and follow fertility policy are enrolled by health workers at the family planning services center [27,28]. Sociodemographic characteristics of both spouses, including age, ethnicity, education level, occupation, place of residence, household registration, and pregnancy history, were obtained face-to-face by trained health workers using

different standardized questionnaires. All information obtained was then uploaded to a specialized health services information system [7].

**Measures.** The NFPHEP provided free syphilis screening test for both spouses. Blood samples (5 mL) were collected by trained and qualified nurses after a fast of at least 8 hours. Then samples were stored at 4–8˚C and transported for analysis within 24 hours [7,28]. Syphilis serology testing was performed by the RPR test in local laboratories affiliated with medical institutions under qualified quality control mechanisms. Test kits approved by the National Medical Products Administration were selected by local laboratories according to their preferences. As a national project, quality control was performed regularly with the following control measures: a) establish the commission of the National Center of Clinical Laboratories for Quality Inspection and Detection, including experts from chemical, clinical, microbiological, and immunological laboratories; b) convene of committees to discuss critical factors that may affect laboratory results and draft quality control plans for key items; c) conduct external quality assessments twice a year, covering 13 items (including the results of the RPR test). At each randomly selected site, five samples were randomly selected. The scoring system was as follows: five correct was 100 points, four correct was 80 points, and less than four correct was unqualified [7,28]. Scores ≥80 were scaled as excellent. The excellent rate was reported to be over 96% [29]. If the result was positive, the health workers would inform the person and their spouses and refer them to a specialist for further confirmation and treatment.

The primary outcome of this study was PSS. We computed temporally smoothed rates as the sum of seropositive cases divided by the sum of participants across the years and then multiplied by 100,000. The temporally smoothed rates were computed because they provide a more representative measure of the burden of diseases over a long period, compared with the burden recorded over a one-year period [20].

**Covariates.** In line with previous studies [8,30], the following potential confounders were extracted or calculated from the baseline information of participants: gender (male or female); ethnicity (han or minority); educational level (primary school or below, junior high school, senior high school, or college and above); occupation (farmer, worker, businessman, service personnel, civil servant, or others); household registration (rural or urban); smoking (no or yes); drinking (no or yes); ever used drugs (no or yes), syphilis seropositivity of spouses (negative or positive) region (Pearl River Delta, East Wing, West Wing, or Mountainous Area) and previous pregnancy (no or yes, only for female). The variable "syphilis seropositivity of spouses" refers to the test results of RPR of their spouses, and a positive result means either a historical or an ongoing syphilis infection [12]. And participants were judged to be migrants if they were enrolled in a municipality different from that in which they were registered, and vice versa. In this study, DoM referred to the time difference (in years) between syphilis screening and marriage registration.

## Statistical analysis

We used numbers and proportions to assess the distribution of sociodemographic characteristics according to age groups, and categorical variables were tested using chi-square tests. We used the median and interquartile range (IQR) to describe the DoM and age of married individuals. PSS for different subgroups of sociodemographic characteristics during 2014–2019 with corresponding 95% confidence intervals (CIs) were calculated. To visualize the temporal changes in PSS from 2014 to 2019, temporal curves of monthly detection rates were plotted. We used geographic information systems (GIS) to illustrate the spatial distribution. Additionally, we performed hot spot analysis (HSA) using the Getis-Ord Gi* statistic with first order Queen contiguity weights to identify county-level hot spots [20]. A hot spot is a specific

geographic unit that has a statistically significantly higher value and is also surrounded by neighboring units that have significantly higher values, while a cold spot is the inverse [20]. We used GeoDa version 1.20 to perform HSA and ArcMap version 10.2 to map.

We plotted scatter plots of DoM and age against PSS and fitted curves with third-order polynomials. Since syphilis infection in both spouses was not independent of each other, the following models were all stratified by gender. We first categorized age (21–24, 25–29, 30–34, 35–39, 40–44, or 45–49) and DoM (0, 0.1–1.0, or >1.0 years) based on the distribution of exposures. DoM of no more than 18 days was the first subgroup (0 year), and DoM between 19 days and 365 days was considered to be "0.1–1.0 years". Univariate logistic regression models were fitted for DoM and age, and then multivariate models were used to control the influence of potential confounders: ethnicity, educational, occupation, household registration, migrant, smoking, drinking, ever used drug, syphilis seropositivity of spouses, region, and pregnancy history (only for women). Complete cases were used. We calculated a Spearman correlation between DoM and age. To examine the independent association of DoM and age with PSS, we additionally adjusted for DoM or age in the above multivariate models. We used the variance inflation factor (VIF) <10 to control the potential multicollinearity problem. We tested for trend by treating the categorical DoM and age as continuous variables in the models.

We then used the restricted cubic spline (RCS) based on multivariate logistic regression to flexibly model the association of DoM and age with PSS. In the spline models, DoM and age were treated as continuous variables, and the reference values were set at their minimum values. Based on the minimum Akaike Information Criterion (AIC), we chose the 5-knot RCS [16]. Because the position of knots has a little impact on the shape of dose-response association [16], we chose the default setting of the "rms" package in R version 4.1.2. To test potential non-linearity, we compared the model with a linear term to another model with its splined model by analysis of variance (ANOVA) test. We further applied the piecewise multivariate logistic regression model to calculate adjusted odds ratios (aOR) per unit increase in DoM and age to examine the threshold effect [31]. The threshold was selected by moving along a pre-defined interval and detecting the inflection point that gave the minimum AIC [32].

We evaluated whether the combined effects of age and DoM on PSS were more or less than additive, which was considered to be most appropriate for assessing the public health importance of interactions [33]. The possible additive interaction was measured by relative excess risk due to interaction (RERI), attributable proportion (AP), and synergy index (SI) [34]. When RERI and AP were greater than 0, and SI was greater than 1, it denoted a synergetic interaction; when RERI and AP were equal to 0, and SI was equal to 1, we considered the absence of additive interaction. In order to present the magnitude of the joint associations in a more straightforward manner, we used the threshold of age identified in the above procedure as a cut off to classify age into a binary variable (< threshold and ≥ threshold). DoM was also dichotomized using the median value (0.2 years) as the cut-point. Then a new variable was created to represent the combination of age and DoM and could be classified into four categories: age < threshold and DoM < 0.2 years, age < threshold and DoM ≥ 0.2 years, age ≥ threshold and DoM < 0.2 years, and age ≥ threshold and DoM ≥ 0.2 years. We chose the stratum with the lowest risk as the reference category [35].

We also performed several stratified analyses to investigate whether the association of DoM and age with PSS varied across calendar year, region, and whether the population was migrant. Since the missing data for occupation and education level were relatively high (**Table 1**), we conducted sensitivity analyses with no adjustment for occupation, education level, or both. We considered a two-sided test $P < 0.05$ to be statistical significance. All analyses were performed with R version 4.1.2.

**Table 1. Sociodemographic characteristics of married individuals aged 21–49 who participated in NFPHEP in Guangdong, 2014–2019.**

| Characteristics | Overall (N = 4,826,214) [#] | Male (N = 2,398,914) [#] | Female (N = 2,427,300) [#] |
|---|---|---|---|
| **Age**[*] | | | |
| 21–24 | 943,829 (19.6) | 309,162 (12.9) | 634,667 (26.1) |
| 25–29 | 2,186,185 (45.3) | 1,085,449 (45.2) | 1,100,736 (45.3) |
| 30–34 | 1,051,497 (21.8) | 619,244 (25.8) | 432,253 (17.8) |
| 35–39 | 425,198 (8.8) | 245,275 (10.2) | 179,923 (7.4) |
| 40–44 | 167,472 (3.5) | 104,136 (4.3) | 63,336 (2.6) |
| 45–49 | 52,033 (1.1) | 35,648 (1.5) | 16,385 (0.7) |
| **Ethnicity**[*] | | | |
| Han | 4,589,320 (95.1) | 2,283,321 (95.2) | 2,305,999 (95.0) |
| Minority | 39,728 (0.8) | 17,542 (0.7) | 22,186 (0.9) |
| Missing | 197,166 (4.1) | 98,051 (4.1) | 99,115 (4.1) |
| **Education**[*] | | | |
| Primary school or below | 95,406 (2.0) | 45,253 (1.9) | 50,153 (2.1) |
| Junior high school | 1,328,985 (27.5) | 652,595 (27.2) | 676,390 (27.9) |
| Senior high school | 1,176,472 (24.4) | 604,513 (25.2) | 571,959 (23.6) |
| College or above | 1,628,103 (33.7) | 801,227 (33.4) | 826,876 (34.1) |
| Missing | 597,248 (12.4) | 295,326 (12.3) | 301,922 (12.4) |
| **Occupation**[*] | | | |
| Farmer | 969,280 (20.1) | 467,522 (19.5) | 501,758 (20.7) |
| Workers | 1,147,034 (23.8) | 606,236 (25.3) | 540,798 (22.3) |
| Businessman | 297,340 (6.2) | 191,931 (8.0) | 105,409 (4.3) |
| Others | 1,659,315 (34.4) | 763,066 (31.8) | 896,249 (36.9) |
| Missing | 753,245 (15.6) | 370,159 (15.4) | 383,086 (15.8) |
| **Household registration**[*] | | | |
| Rural | 3,512,075 (72.8) | 1,720,175 (71.7) | 1,791,900 (73.8) |
| Urban | 1,309,574 (27.1) | 676,229 (28.2) | 633,345 (26.1) |
| Missing | 4,565 (0.1) | 2,510 (0.1) | 2,055 (0.1) |
| **Migrant population**[*] | | | |
| No | 4,032,434 (83.6) | 2,044,332 (85.2) | 1,988,102 (81.9) |
| Yes | 792,265 (16.4) | 353,635 (14.7) | 438,630 (18.1) |
| Missing | 1,515 (0.0) | 947 (0.0) | 568 (0.0) |
| **Cigarette smoking**[*] | | | |
| No | 4,126,965 (85.5) | 1,729,781 (72.1) | 2,397,184 (98.8) |
| Yes | 661,804 (13.7) | 652,659 (27.2) | 9,145 (0.4) |
| Missing | 37,445 (0.8) | 16,474 (0.7) | 20,971 (0.9) |
| **Alcohol drinking**[*] | | | |
| No | 3,734,558 (77.4) | 1,529,069 (63.7) | 2,205,489 (90.9) |
| Yes | 1,049,885 (21.8) | 852,578 (35.5) | 197,307 (8.1) |
| Missing | 41,771 (0.9) | 17,267 (0.7) | 24,504 (1.0) |
| **Ever used drugs**[*] | | | |
| No | 4,774,080 (98.9) | 2,374,988 (99.0) | 2,399,092 (98.8) |
| Yes | 773 (0.0) | 587 (0.0) | 186 (0.0) |
| Missing | 51,361 (1.1) | 23,339 (1.0) | 28,022 (1.2) |
| **Syphilis seropositivity of spouses**[*] | | | |
| Negative | 4,662,448 (96.6) | 2,351,609 (98.0) | 2,310,839 (95.2) |
| Positive | 12,297 (0.3) | 6,341 (0.3) | 5,956 (0.2) |
| Missing | 151,469 (3.1) | 40,964 (1.7) | 110,505 (4.6) |

(*Continued*)

**Table 1.** (Continued)

| Characteristics | Overall (N = 4,826,214) [#] | Male (N = 2,398,914) [#] | Female (N = 2,427,300) [#] |
|---|---|---|---|
| **Duration of marriage (year)**[*] | | | |
| 0 | 1,566,318 (32.5) | 798,613 (33.3) | 767,705 (31.6) |
| 0.1–1.0 | 1,112,045 (23.0) | 555,203 (23.1) | 556,842 (22.9) |
| >1.0 | 1,471,135 (30.5) | 708,262 (29.5) | 762,873 (31.4) |
| Missing | 676,716 (14.0) | 336,836 (14.0) | 339,880 (14.0) |
| **Previous pregnancy (Only female)** | | | |
| No | 1,451,616 (59.8) | - | 1,451,616 (59.8) |
| Yes | 962,304 (39.6) | - | 962,304 (39.6) |
| Missing | 13,380 (0.6) | - | 13,380 (0.6) |

Notes: Duration of marriage referred to the time difference (in years) between syphilis screening and marriage registration. Participants in the first subgroup (0 year) are newly married couples, and the duration of marriage was no more than 18 days. Moreover, the duration of marriage between 19 days and 365 days was considered to be "0.1–1.0 years".

Abbreviations: NFPHEP, National Free Preconception Health Examination Project; *CI*, confidence interval.

[#] Data was presented as No. (%); reported percentages are composition ratios of each horizontal item.

[*] $P < 0.05$

## Result

### Study participants

From January 1, 2014 to November 10, 2019, a total of 5,395,970 married individuals were enrolled in the NFPHEP in Guangdong and 4,826,214 (89.4%) participants were included in the analyses (**S1 Fig**). The median age of men was 29 (IQR: 26–32) years, and the median DoM was 0.2 (IQR: 0–2.5) years. For women, the median age was 27 (IQR: 24–30) years and the median DoM was 0.2 (IQR: 0–2.9) years. Baseline characteristics of all participants were reported in **Table 1** and the PSS of subgroups was showed in **S1 Table**. Among them, 2,427,300 (50.7%) were female; 2,186,185 (45.3%) were 25–29 years old; 4,589,320 (95.1%) were of Han nationality; 2,804,575 (58.1%) had an education level higher than junior high school; 792,265 (16.4%) were migrant; and 2,406,696 (49.9%) came from the Pearl River Delta region. And 2,678,363 (55.5%) were married for less than one year. In addition, **S2 and S3 Tables** show the baseline characteristics according to age categories by gender.

### Spatial and temporal distribution of syphilis

In total, 12,867 syphilis cases were detected among married individuals aged 21–49 years in Guangdong Province during 2014–2019, with a PSS of 266.61 (95% CI: 262.03–271.24) per 100,000 (**S1 Table**). The PSS of 254.70 (95% CI: 248.37–261.15) in married men was significantly lower than that of 278.38 (95% CI: 271.79–285.07) in married women ($P <0.001$). **Fig 2** showed the monthly change in PSS and the annual PSS for men decreased from 2014 (292.54 per 100,000) to 2019 (240.14 per 100,000), while the PSS for women remained largely unchanged (**S4 Table**). **Fig 3** illustrated the spatial heterogeneity of PSS among married individuals aged 21–49 years in Guangdong Province. Specifically, married individuals residing in Mountainous Area had the highest PSS (294.25 per 100,000), whereas the lowest PSS was observed in the Pearl River Delta (254.42 per 100,000) (**Fig 3 and S1 Table**). Notably, 47.6% of positive cases came from the Pearl River Delta region (**S1 Table**). The HAS showed there were six significant syphilis hotspot counties: Xinxin District, Yingde City, Fogang County, Yuexiu District, Panyu District, and Jiexi County (**Fig 3**).

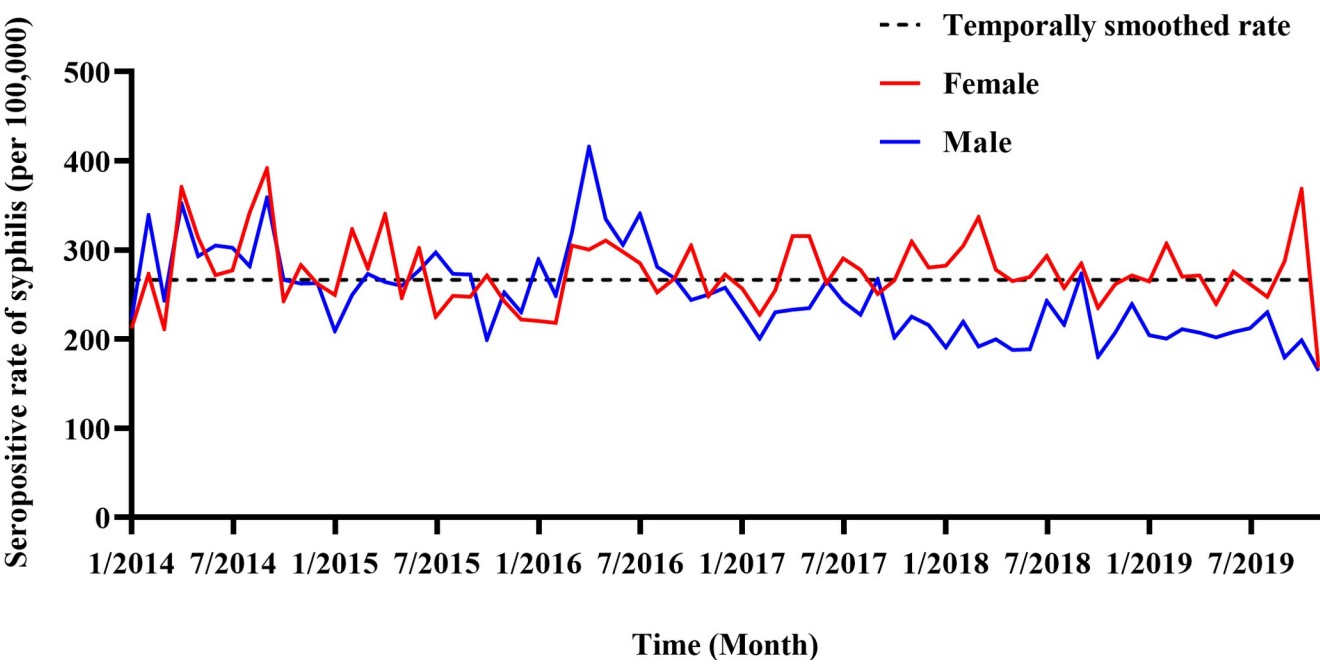

**Fig 2. Preconception syphilis seroprevalence among married individuals aged 21–49 years enrolled in the National Free Preconception Health Examination Project in Guangdong Province, 2014–2019.** We computed temporally smoothed rates as the sum of all seropositive cases divided by the sum of participants across the years, and then multiplied by 100,000.

## Association of DoM and age with PSS

The Spearman correlation between DoM and age was 0.49 in males and 0.53 in females. In the scatter plot, we observed that age showed a strong J-shaped association with PSS, and PSS slowly increased with increasing DoM (**S2 Fig**). The multivariable adjusted model showed a

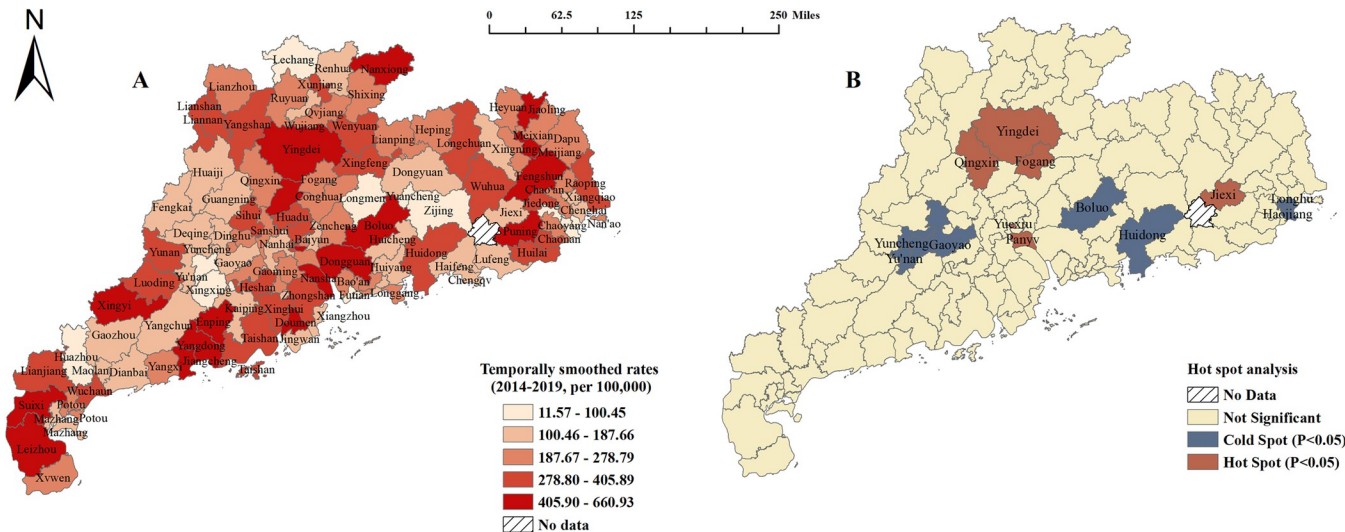

**Fig 3. Spatial distribution of preconception syphilis seroprevalence among married individuals aged 21–49 years, 2014–2019.** Base layer of the map was downloaded from Resource and Environment Science and Data Center (https://www.resdc.cn/data.aspx?DATAID=201). A: Distribution of syphilis at the county level in Guangdong Province. B: Hot spot analysis result of the county-level syphilis seroprevalence within 122 contiguous counties in Guangdong Province. We calculated temporally smoothed rates as the sum of all seropositive cases divided by the sum of participants over the period 2014–2019, multiplied by 100,000. No data were available for Luhe county and no neighboring counties in Nanao county and they were excluded from the hot spot analysis. Note: The hot spots were determined using Getis-Ord Gi* statistic with first order Queen contiguity weights.

**Table 2. Odds ratio (95% CI) of PSS according to duration of marriage and age among married individuals aged 21–49 in Guangdong Province, 2014–2019.**

| Characteristic | Cases | PSS per 100,000 (95% CI) | cOR (95% CI) | Model 1* (95% CI) | Model 2# (95% CI) |
|---|---|---|---|---|---|
| **Male** | | | | | |
| **Age (years)** | | | | | |
| 21–24 | 531 | 171.75 (157.48–186.98) | 1.00 (reference) | 1.00 (reference) | 1.00 (reference) |
| 25–29 | 1728 | 159.20 (151.79–166.87) | 0.93 (0.84–1.02) | 1.06 (0.95–1.18) | 1.07 (0.96–1.20) |
| 30–34 | 1394 | 225.11 (213.47–237.22) | 1.31 (1.19–1.45) | 1.55 (1.38–1.74) | 1.66 (1.47–1.88) |
| 35–39 | 1247 | 508.41 (480.71–537.27) | 2.97 (2.68–3.29) | 3.53 (3.14–3.96) | 3.94 (3.47–4.49) |
| 40–44 | 847 | 813.36 (759.70–869.78) | 4.77 (4.28–5.31) | 5.69 (5.01–6.45) | 6.58 (5.72–7.57) |
| 45–49 | 363 | 1018.29 (916.67–1128.00) | 5.98 (5.23–6.84) | 6.92 (5.92–8.07) | 7.94 (6.70–9.40) |
| P value for trend | - | - | < 0.001 | < 0.001 | < 0.001 |
| **Duration of marriage (years)** | | | | | |
| 0 | 3833 | 244.71 (237.05–252.57) | 1.00 (reference) | 1.00 (reference) | 1.00 (reference) |
| 0.1–1.0 | 2645 | 237.85 (228.89–247.07) | 1.06 (0.98–1.14) | 1.04 (0.96–1.13) | 1.04 (0.96–1.13) |
| > 1.0 | 4523 | 307.45 (298.58–316.51) | 1.52 (1.42–1.61) | 1.42 (1.32–1.52) | 0.83 (0.77–0.90) |
| P value for trend | - | - | < 0.001 | < 0.001 | < 0.001 |
| **Female** | | | | | |
| **Age (years)** | | | | | |
| 21–24 | 1623 | 255.72 (243.46–268.44) | 1.00 (reference) | 1.00 (reference) | 1.00 (reference) |
| 25–29 | 2362 | 214.58 (206.03–223.40) | 0.84 (0.79–0.89) | 0.99 (0.92–1.07) | 1.05 (0.97–1.14) |
| 30–34 | 1360 | 314.63 (298.18–331.75) | 1.23 (1.15–1.32) | 1.45 (1.33–1.59) | 1.72 (1.56–1.90) |
| 35–39 | 898 | 499.10 (467.06–532.75) | 1.96 (1.80–2.12) | 2.38 (2.14–2.64) | 3.02 (2.70–3.38) |
| 40–44 | 396 | 625.24 (565.33–689.73) | 2.45 (2.20–2.74) | 2.67 (2.31–3.08) | 3.46 (2.97–4.02) |
| 45–49 | 118 | 720.17 (596.46–861.83) | 2.83 (2.35–3.41) | 2.36 (1.79–3.10) | 3.11 (2.33–4.16) |
| P value for trend | - | - | < 0.001 | < 0.001 | < 0.001 |
| **Duration of marriage (years)** | | | | | |
| 0 | 2138 | 278.49 (266.84–290.52) | 1.00 (reference) | 1.00 (reference) | 1.00 (reference) |
| 0.1–1.0 | 1398 | 251.06 (238.10–264.54) | 0.90 (0.84–0.96) | 0.86 (0.79–0.93) | 0.87 (0.81–0.94) |
| > 1.0 | 2246 | 294.41 (282.40–306.81) | 1.06 (1.00–1.12) | 0.72 (0.67–0.79) | 0.66 (0.61–0.71) |
| P value for trend | - | - | 0.062 | < 0.001 | < 0.001 |

Abbreviations: CI, confidence interval; PSS, preconception syphilis seroprevalence; cOR, crude odds ratio.

Duration of marriage referred to the time difference (in years) between syphilis screening and marriage registration. Participants in the first subgroup (0 year) are newly married couples, and the duration of marriage was no more than 18 days. Moreover, the duration of marriage between 19 days and 365 days was considered to be "0.1–1.0 years".

* Adjusted for ethnicity, education, occupation, household registration, migrant, smoking, drinking, ever used drug, syphilis seropositivity of spouses and region. Additional adjustment of pregnancy history for women.

# On the basis of model 1, additionally, adjusted for duration of marriage or age.

positive association between age and PSS in both males and females (**Table 2**). DoM was negatively associated with PSS in females, whereas it was positively associated among males. In the mutually adjusted model including DoM and age, we continued to observe a J-shaped positive association between age and PSS, while among males DoM turned out to be a protective factor. The aOR for syphilis seropositivity was 3.11 (95% CI: 2.33–4.16) for married women aged 45–49 years compared to married women aged 21–24 years. And married men aged 45–49 years had an aOR of 7.94 (95% CI: 6.70–9.40) compared to married men aged 21–24 years.

The results of RCS also showed a strong J-shaped association between age and PSS in married individuals aged 21–49 years (P for non-linearity <0.001) (**Fig 4**). The relative risk of

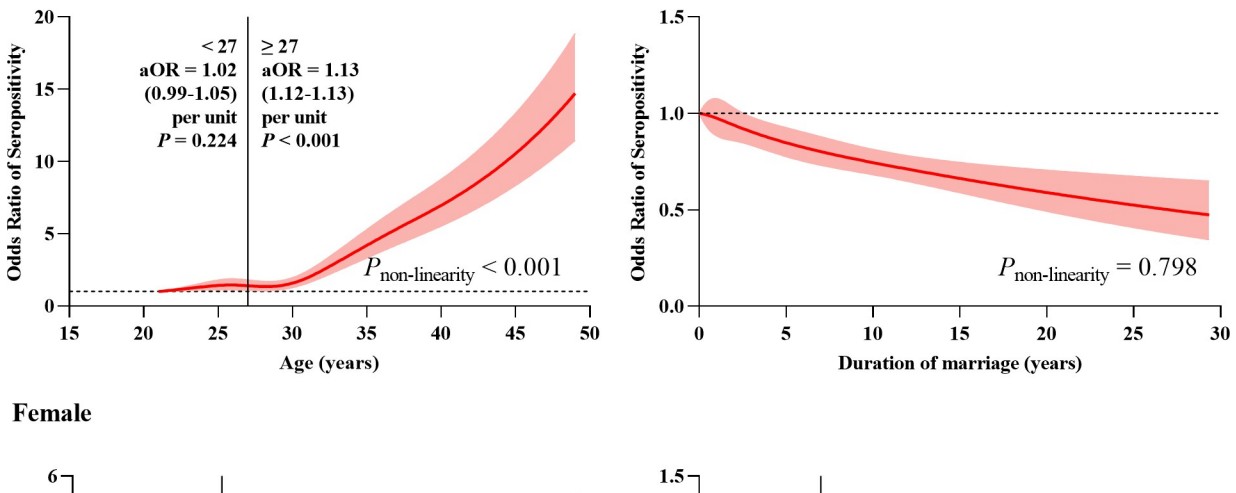

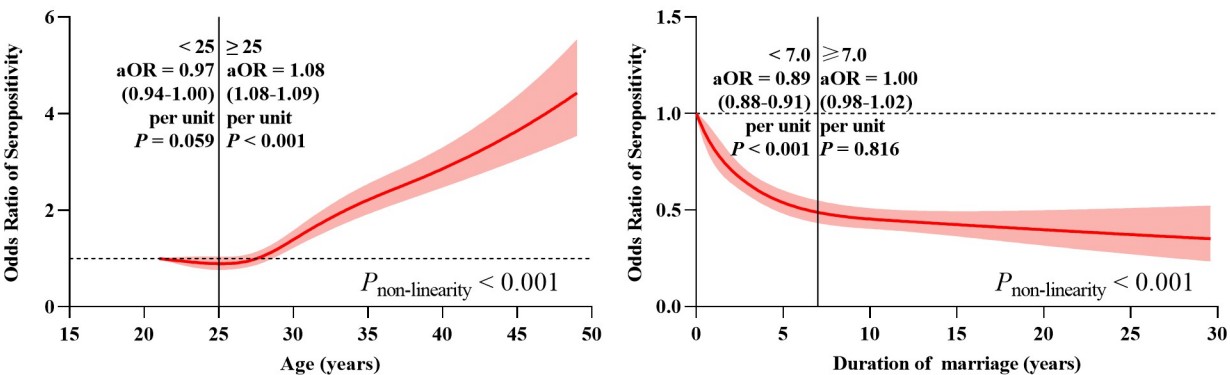

**Fig 4. Association of duration of marriage and age with preconception syphilis seroprevalence among married individuals aged 21–49 years.** Adjusted odds ratios (aOR) were indicated by solid lines and 95% confidence intervals by shaded areas. Duration of marriage and age were mutually adjusted in the restricted cubic splines models with five knots and the reference points were the respective minimum values. All models were adjusted for ethnicity, education, occupation, household registration, migrant, smoking, drinking, ever used drug, syphilis seropositivity of spouses and region. Additional adjustment of pregnancy history only for women.

syphilis seropositivity was relatively flat until age 27 for males (age 25 for females) and increased rapidly afterwards (*P* for non-linearity <0.001). Among married men older than 27 years, the aOR was 1.13 (95% CI: 1.12–1.13, *P* <0.001) per unit, compared with 1.02 (95% CI: 0.99–1.05, *P* = 0.224) for those younger than 27 years. Among married women older than 25 years, the aOR was 1.08 (95% CI: 1.08–1.09, *P* <0.001) per unit, compared with 0.97 (95% CI: 0.94–1.00, *P* = 0.059) for those younger than 25 years. DoM was linearly associated with PSS in men (*P* for non-linearity = 0.798), whereas women showed a nonlinear association (*P* for non-linearity <0.001). In married women, the risk of syphilis seropositivity first decreased rapidly with increasing DoM and then decreased slowly above 7 years.

Our findings remained robust in sensitivity analyses (**S3 Fig**). The results did not change with no adjustment for occupation, or education level or both. We also examine the associations stratified by calendar year, region, and migrant status (**S4, S5, and S6 Figs**). In 2014 and 2015, the association between DoM and PSS was not monotonical (**S4 Fig**). We observed a weaker J-shaped association between age and PSS among married women in the East Wing than in other areas (**S5 Fig**). The association of DoM and age with PSS was stronger among migrant married women. In contrast, the association between age and PSS was stronger among non-migrant married men (**S6 Fig**).

**Table 3. The interactive effects between age and DoM on PSS among married individuals aged 21–49 years.**

| Age (years) | DoM (years) | Total no. | No. of cases | aOR* (95% CI) | P |
|---|---|---|---|---|---|
| Male[#] | | | | | |
| < 27 | < 0.2 | 420,884 | 649 | 1.00 (reference) | - |
| < 27 | ≥ 0.2 | 187,732 | 316 | 1.03 (0.89–1.20) | 0.677 |
| ≥ 27 | < 0.2 | 576,388 | 1,498 | 1.79 (1.62–1.97) | < 0.001 |
| ≥ 27 | ≥ 0.2 | 877,074 | 2,756 | 2.09 (1.91–2.30) | < 0.001 |
| RERI | 0.27 (95% CI: 0.08–0.47) | | | | |
| AP | 0.13 (95% CI: 0.04–0.23) | | | | |
| SI | 1.34 (95% CI: 1.04–1.71) | | | | |
| Female | | | | | |
| < 25 | ≥ 0.2 | 170,404 | 465 | 1.00 (reference) | - |
| < 25 | < 0.2 | 369,191 | 915 | 1.17 (1.03–1.33) | 0.020 |
| ≥ 25 | ≥ 0.2 | 953,337 | 2,686 | 1.14 (1.01–1.28) | 0.035 |
| ≥ 25 | < 0.2 | 594,488 | 1,716 | 1.59 (1.41–1.80) | < 0.001 |
| RERI | 0.29 (95% CI: 0.14–0.44) | | | | |
| AP | 0.18 (95% CI: 0.08–0.28) | | | | |
| SI | 1.95 (95% CI: 1.04–3.65) | | | | |

Abbreviations: DoM, duration of marriage; PSS, preconception seroprevalence of syphilis; No., number; CI, confidence interval; aOR, adjusted odds ratio; RERI: Relative Excess Risk due to Interaction; AP, attributable proportion; SI; synergy index.

* All models adjusted for ethnicity, education, occupation, household registration, migrant, smoking, drinking, ever used drug, syphilis seropositivity of spouses and region. Additional adjustment of pregnancy history for women.

[#] Among married men, dichotomous DoM was a protective factor, so we chose the stratum with the lowest risk as the reference category.

## Interactive effects of DoM and age on PSS

Table 3 showed the interaction between DoM and age on PSS among married individuals aged 21–49 years. Among married men, dichotomous DoM was a protective factor, so we chose the stratum with the lowest risk as the reference category (i.e., age < 27 years and DoM < 0.2 years), and estimates suggested a synergistic effect of age ≥ 27 years and DoM ≥ 0.2 years (RERI > 0, AP > 0, and SI > 1). Among married women, we found a synergistic effect of age ≥ 25 years and DoM < 0.2 years, with the RERI from adjusted models of 0.29 (95% CI: 0.14–0.44), suggesting 29% excess risks relative to expectations based on the independent effects estimated for each exposure alone.

## Discussion

Using data from over 4.8 million married individuals aged 21–49 years old in Guangdong Province, we found that there was a J-shaped non-linearity association, rather than a linear relationship, between age and PSS. Specifically, the risk of syphilis seropositivity increased rapidly with age among married women over the age of 25 and married men over the age of 27. Moreover, results showed that DoM was negatively associated with PSS for both sexes. These results suggest that in order to eliminate MTCT, screening for syphilis should be recommended in those who were newly married and in their late 20s and older, and early prevention strategies should be implemented in young people as early as possible.

The PSS of married individuals aged 21–49 years in Guangdong Province was 266.61 per 100,000 persons, consistent with previously reported rates among this population in China for the period 2013–2018 [7], with a reported PSS of 0.30% among men and 0.38% among women. Compared to a similar study conducted in rural Guangdong in 2011 [36], the

decreased in PSS was probably due to active and effective prevention and control measures [25]. Moreover, we observed a regional variation. The differences in the economy, population composition, and meteorologic conditions may contribute to these variations [23]. We found that less developed areas have higher PSS, which was consistent with the national report and reports from other low-income and middle-income countries [7,37]. This may be due to insufficient health resources [7]. This suggests that in resource-limited areas, guaranteeing the support in terms of financial subsidy, clinical professional training, disease surveillance, and screening of high-risk groups might make sustainable progress in the prevention of MTCT of syphilis [25].

Consistent with previous studies [4,8], the findings of this study showed that age was positively associated with PSS. Since the RPR test cannot distinguish historical or current infections, cumulative effect did exist [8]. Both the newly infected syphilis and the treated syphilis in pregnant women should be identified and well-managed [3], because even after treatment, the risk of adverse pregnancy outcomes remains significantly higher compared to uninfected pregnancies [3]. In addition, the reinfection rate of syphilis is high [38], with 10.94% reported in Shenzhen [39], 9.17% reported in Belgium [40], and 7.3% reported in Florida [41]. Thus it was recommended that pregnant women with previous syphilis infection should be screened again at 28 weeks of gestation and at delivery [12]. Considering that during the period of the universal two-child policy, the proportion of births to women older than 35 years increased from 8.52% in 2013 to 15.82% in 2017 in China [42], more attention should be paid to routine screening for syphilis. This also suggests that syphilis prevention education in the early stages of adulthood may effectively reduce the burden of syphilis among married individuals, which in turn may help to eliminate MTCT. Moreover, we found a stronger J-shaped association between age and PSS among married men than among married women. This could due to men being more likely to have an affair [43], which has been linked to an increased risk of syphilis infection.

For the association of DoM with PSS in males, we obtained opposite results in the two models without and with adjustment for age. This may be due to the fact that age was an important confounding factor in the association of DoM and PSS in males. The Spearman correlation shows that age and DoM is positively correlated. Moreover, age was strongly associated with PSS. Therefore, in model 1, without adjustment for age, it showed that DoM was a risk factor. The results for model 1 and 2 were similar in females, which may be due to the slightly weaker association between age and PSS in females compared to males. In summary, we found a negative association between DoM and PSS, and this association was stronger among females. With a dual-standard about sexual behavior for men and women, many of societies found women most often infected by their husbands [44]. For example, a study of 26,230 couples in India found that 75% of couples with syphilis infection were introduced by husbands [45]. This gives circumstantial evidence of the important role of male sexual activity in the MTCT of syphilis. The CDC regarded sexual partner infection as an important factor for syphilis prevention [7]. We found older non-migrant married men were a high-risk population for syphilis transmission, consistent with a population-based study of chlamydia in China [44]. Because these people seemed to be highly paid and were more likely to engage in unprotected commercial sex [44]. All of these emphasize that in areas with a heavy burden of MTCT, integrated syphilis screening for couples should be considered for maternal care to eliminate the risk of MTCT [7].

We also assessed the additive-scale interactions between DoM and age on PSS. Among married men, we found DoM was positively associated with PSS after dichotomizing at the median value, which runs counter to the finding of RCS. This opposite result may be due to the loss of information after categorizing continuous variables [15]. Among married women,

the older women (> 25 years old) with shorter DoM (< 0.2 years) was more likely to be sero-positive. This was in line with the research conducted in rural Malawi, where being in a stable marriage appears to be a protective factor against HIV infection among young women [46]. Because pre-marital sexual activity seems to be associated with a higher risk of being infected with sexually transmitted diseases (STD) [18].

MTCT for syphilis is a solvable public health problem [47], but critical details about prevention and control strategies are needed for successful implementation. When deciding who to screen for syphilis, clinicians should know the prevalence of syphilis infection in the communities they serve and other sociodemographic factors that may be associated with an increased risk of syphilis infection [11]. In terms of resource allocation, attention should be paid to preventing MTCT of syphilis in underdeveloped areas, and efforts should be made to have 95% of pregnant women tested for syphilis during pregnancy [47]. In clinical practice, screening for syphilis in newly married individuals who are in their late 20s or older should be recommended. Medical professionals could advise pregnant women's old male spouses to undergo syphilis testing as well. Educating people in early adulthood about syphilis prevention as early as possible will not only increase their awareness of using condoms, avoiding high-risk sexual partners and taking other protective measures, but also mitigate the increasing cost of syphilis to the health system [48]. Integrated syphilis screening for couples should also be considered to eliminate the risk of MTCT. Furthermore, the control of syphilis or other STDs is not only a medical issue but also a social issue because of the related stigma and discrimination [49]. Public service adverts, leaflets, postcards, and posters can be used to increase awareness of syphilis among the public [48].

## Strengths and limitations

Our study has several strengths. First, we did a population-based design, which is considered the gold standard in evaluating disease epidemics [7]. Surveillance data imperfectly indicates current prevalence and risk factors because asymptomatic infection, incomplete coverage, and other issues could lead to underreporting [44]. Second, the study sample included over 4.8 million married individuals of reproductive age spanning 6 years, which enabled us to explore the association with sufficient statistical power. Third, the NFPHEP was well established and had standard quality control, and the data obtained by trained personnel was reliable. Fourth, we reported the interaction on the additive scale, which is useful in targeting specific populations and in terms of resource allocation [33].

The study also has several limitations. First, syphilis screening in the NFPHEP was performed with the RPR test. The sensitivity of the RPR test is 86%, 100%, and 98% for primary, secondary, and latent syphilis, respectively, and the specificity is about 98% [50]. A study in the Greater Toronto Area showed that 9,377 (77.2%) were confirmatory positive out of 12,140 RPR positive samples, based on 2,055,913 samples [51]. It is quick (provides results within 20 minutes), simple and inexpensive and is recommended by the World Health Organization (WHO) in resource-limited settings [14]. However, concerns regarding serological false positives and false negatives could still be raised. Further well-designed studies, using the same brands of diagnostic kits and the same laboratory teams, are needed to delimit cases of false negatives and positives. Second, only the operational data was used in the analyses, but with no sexual behaviors. Therefore, we cannot entirely rule out the potential residual confounding caused by unmeasured or unknown factors. Third, this study was a cross-sectional study, which limits our ability to make causal inference. Fourth, the study participants were enrolled by health workers and restricted to married individuals planning a pregnancy within the next six months. Married individuals who did not plan to have a baby during the study period were

excluded, which may lead to the selection bias. The NFPHEP was well implemented in Guang-dong Province, with a coverage rate of over 85% of the target population [24]. Preconception health care is also recommended in the United States, while the preconception health care rate is about 50% [52]. We believe that our main findings have broader implications within China and for other areas of the heavy burden of MTCT.

## Conclusions

In summary, our results provided new empirical evidence for the association of DoM and age with PSS and new insights into the early prevention and control of MTCT. Individuals living in underdeveloped areas, newly married, and in their late 20s or older have higher PSS. In terms of resource allocation, attention should be paid to preventing MTCT of syphilis in underdeveloped areas. Screen for syphilis in newly married individuals who are in their late 20s or older should be recommended. Medical professionals could advise pregnant women's male spouses to undergo syphilis testing as well. Additionally, early syphilis prevention strategies should be implemented among young people as early as possible to mitigate the increasing cost of syphilis to the health system.

## Supporting information

**S1 STROBE Checklist. Checklist of items for observational studies.**
(DOCX)

**S1 Table. Sociodemographic characteristics and seroprevalence of syphilis among married individuals aged 21–49 who participated in NFPHEP in Guangdong, 2014–2019.**
(DOCX)

**S2 Table. Characteristics of married female aged 21–49 who participated in NFPHEP in Guangdong, according to age categories 2014–2019.**
(DOCX)

**S3 Table. Characteristics of married male aged 21–49 who participated in NFPHEP in Guangdong Province during 2014–2019, according to age categories.**
(DOCX)

**S4 Table. Preconception syphilis seroprevalence among married individuals aged 21–49 years by calendar year.**
(DOCX)

**S1 Fig. Flow chart of participants in the National Free Preconception Health Screening Project (NFPHEP) in Guangdong Province during 2014–2019.**
(TIF)

**S2 Fig. Scatter plots of duration of marriage and age against preconception syphilis sero-prevalence.** Seroprevalence of syphilis was calculated as the sum of all seropositive cases divided by the sum of participants over the period 2014–2019, multiplied by 100,000. The curve is fitted by a third-order polynomial.
(TIF)

**S3 Fig. Sensitivity analysis of duration of marriage and age in relation to preconception syphilis seropositivity among married individuals aged 21–49 years.** Adjusted for ethnicity, education, occupation, household registration, migrant, smoking, drinking, ever used drug, syphilis seropositivity of spouses and region and pregnancy history (only for women). Dura-tion of marriage and age were mutually adjusted in the restricted cubic splines models with

five knots and the reference points were the respective minimum values. Model 1: no adjustment for occupation; Model 2: no adjustment for educational level; Model 3: no adjustment for occupation and educational level.
(TIF)

**S4 Fig. Association of duration of marriage and age with preconception syphilis seropositivity in married individuals aged 21–49 years, stratified by calendar year.** All models were adjusted for ethnicity, education, occupation, household registration, migrant, smoking, drinking, ever used drug, syphilis seropositivity of spouses and region and pregnancy history (only for women). Duration of marriage and age were mutually adjusted in the restricted cubic splines models with five knots and the reference points were the respective minimum values.
(TIF)

**S5 Fig. Association of duration of marriage and age with preconception syphilis seropositivity in married individuals aged 21–49 years, stratified by region.** All models were adjusted for ethnicity, education, occupation, household registration, migrant, smoking, drinking, ever used drug, syphilis seropositivity of spouses and pregnancy history (only for women). Duration of marriage and age were mutually adjusted in the restricted cubic splines models with five knots and the reference points were the respective minimum values.
(TIF)

**S6 Fig. Association of duration of marriage and age with preconception syphilis seropositivity in married individuals aged 21–49 years, stratified by migrant.** All models were adjusted for ethnicity, education, occupation, household registration, smoking, drinking, ever used drug, syphilis seropositivity of spouses and region and pregnancy history (only for women). Duration of marriage and age were mutually adjusted in the restricted cubic splines models with five knots and the reference points were the respective minimum values.
(TIF)

## Acknowledgments

We would like to thank all the community workers, health workers and participants involved in the NFPHEP project for their generous help during the data gathering process.

## Author Contributions

**Conceptualization:** Lu Han, Rui Li, Chaonan Fan, Li Ling.

**Data curation:** Lu Han, Xiaohua Liu, Jiabao Wu, Hua Nie, Weibing Qin.

**Formal analysis:** Wenxue Xiong, Rui Li.

**Funding acquisition:** Lu Han.

**Investigation:** Lu Han, Xiaohua Liu, Jiabao Wu, Hua Nie, Weibing Qin.

**Methodology:** Wenxue Xiong, Lu Han, Rui Li, Xijia Tang, Chaonan Fan, Li Ling.

**Project administration:** Weibing Qin, Li Ling.

**Resources:** Lu Han, Li Ling.

**Software:** Wenxue Xiong, Rui Li.

**Supervision:** Lu Han, Weibing Qin, Li Ling.

**Validation:** Wenxue Xiong, Rui Li.

**Visualization:** Wenxue Xiong.

**Writing – original draft:** Wenxue Xiong.

**Writing – review & editing:** Wenxue Xiong, Lu Han, Rui Li, Xijia Tang, Li Ling.

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
