## [Decision Letter · Decision Letter 0]

28 Jul 2022

Dear Dr. Ling,

Thank you very much for submitting your manuscript "Syphilis seroprevalence and association with duration of marriage and age among married individuals aged 21-49 years in Guangdong Provence, China: A population-based cross-sectional study" for consideration at PLOS Neglected Tropical Diseases. As with all papers reviewed by the journal, your manuscript was reviewed by members of the editorial board and by several independent reviewers. In light of the reviews (below this email), we would like to invite the resubmission of a significantly-revised version that takes into account the reviewers' comments. 

In addition please consider more clearly articulating the clinical importance of your findings given that the key strategy for preventing MTCT of syphilis is antenatal screening of women during pregancy.

We cannot make any decision about publication until we have seen the revised manuscript and your response to the reviewers' comments. Your revised manuscript is also likely to be sent to reviewers for further evaluation.

Sincerely,

Michael Marks

Section Editor

Michael Marks

Section Editor

Reviewer's Responses to Questions

**Key Review Criteria Required for Acceptance?**

**Methods**

-Are the objectives of the study clearly articulated with a clear testable hypothesis stated?

-Is the study design appropriate to address the stated objectives?

-Is the population clearly described and appropriate for the hypothesis being tested?

-Is the sample size sufficient to ensure adequate power to address the hypothesis being tested?

-Were correct statistical analysis used to support conclusions?

-Are there concerns about ethical or regulatory requirements being met?

Reviewer #1: (No Response)

Reviewer #2: (No Response)

**Results**

-Does the analysis presented match the analysis plan?

-Are the results clearly and completely presented?

-Are the figures (Tables, Images) of sufficient quality for clarity?

Reviewer #1: (No Response)

Reviewer #2: (No Response)

**Conclusions**

-Are the conclusions supported by the data presented?

-Are the limitations of analysis clearly described?

-Do the authors discuss how these data can be helpful to advance our understanding of the topic under study?

-Is public health relevance addressed?

Reviewer #1: (No Response)

Reviewer #2: (No Response)

**Editorial and Data Presentation Modifications?**

Reviewer #1: (No Response)

Reviewer #2: (No Response)

**Summary and General Comments**

Reviewer #1: Thank you for the invitation to review this manuscript.

This study has a good idea to describe the seroprevalence and spatiotemporal distribution of syphilis and to investigate the association of duration of marriage (DoM) and age with syphilis seroprevalence (SSP). 

However, there are several minor comments need to be addressed.

Q1 Page 9 Line 134: Information on duration of marriage was collected, it would be better to display the distribution of this variable in the manuscript.

Q2 Page 12 Line 201: To examine the independent association of DoM and age with SSP, why not adjust such potential confounding variables like ethnicity, educational, occupation?

Q3 Page 17 Table 1: The distribution of variables among different age group is showed. Maybe it would be better to demonstrate the total distribution of variables in both sex. Moreover, the distribution of DoM among different age group in specific sex is varied, which could cause potential selection bias to your conclusion made in the same specific sex.

Q4 Page 17 Table 1:The proportion of participants with a 0 year duration of marriage is relatedly large. Does this mean this group of people are all newly married couples and the duration of marriage is less than one month? It is better to be explained in notes.

Q5 Page 30 Line 473: Have you showed the results of subgroup analyses for four regions?

In summary, the article is recommended for publication after minor revisions.

Reviewer #2: This is a population-based epidemiological study focused on reproductive married individuals to find the association of DoM(Duration of marriage) and age with SSP (syphilis seroprevalence). The study is interesting. However, the conclusions remain to be discussed. 

Major revision:

1: The authors used RPR test as screening SSP. How can they avoid the serological false positive or false negative results of RPR? 

2: The main conclusion of this study is that the higher the positive rate of syphilis serum with age. This conclusion may be biased by RPR screening. RPR can be positive for life even after syphilis treatment. Therefore, the newly infected syphilis and the treated syphilis cause the positive rate of RPR to increase with age.

3: In Table 2, the author divided the marriage age into three groups: 0, 0-1.0, >1.0 (years). What does the first group of 0 mean?

4: The authors should explain why in male group, they used two statistical models and got the different results on the relationship between marriage age and RPR positive rate.

PLOS authors have the option to publish the peer review history of their article (what does this mean?). If published, this will include your full peer review and any attached files.

Reviewer #1: No

Reviewer #2: No
---

## [Decision Letter · Decision Letter 1]

11 Sep 2022

Dear Dr. Ling,

Thank you very much for submitting your manuscript "Syphilis seroprevalence and association with duration of marriage and age among married individuals aged 21-49 years in Guangdong Province, China: A population-based cross-sectional study" for consideration at PLOS Neglected Tropical Diseases. As with all papers reviewed by the journal, your manuscript was reviewed by members of the editorial board and by several independent reviewers. In light of the reviews (below this email), we would like to invite the resubmission of a significantly-revised version that takes into account the reviewers' comments. 

We cannot make any decision about publication until we have seen the revised manuscript and your response to the reviewers' comments. Your revised manuscript is also likely to be sent to reviewers for further evaluation.

Sincerely,

Michael Marks

Section Editor

Michael Marks

Section Editor

Reviewer's Responses to Questions

**Key Review Criteria Required for Acceptance?**

**Methods**

-Are the objectives of the study clearly articulated with a clear testable hypothesis stated?

-Is the study design appropriate to address the stated objectives?

-Is the population clearly described and appropriate for the hypothesis being tested?

-Is the sample size sufficient to ensure adequate power to address the hypothesis being tested?

-Were correct statistical analysis used to support conclusions?

-Are there concerns about ethical or regulatory requirements being met?

Reviewer #2: (No Response)

Reviewer #3: 1. Line 154: Are these community workers also considered health workers in the hospital system? Or would these include individuals within the communities/districts? Furthermore, how are couples who are planning on a pregnancy identified (e.g., outreach to districts, through community organizations?)

2. It remains slightly unclear if both spouses were approached to get tested as well as asked to complete separate questionnaires - could this be clarified in the study design?

3. Line 197: For "syphilis seropositivity of spouses" - could the authors clarify if this was a question asking participants about whether they knew if their spouse had syphilis? Furthermore is this a history of Syphilis or ongoing infection? More details on what this means would help to clarify the inclusion of this covariate.

I am unable to assess the robustness of geospatial techniques and analyses mentioned in the methods but I am able to assess the suitability and interpretation of regression models.

**Results**

-Does the analysis presented match the analysis plan?

-Are the results clearly and completely presented?

-Are the figures (Tables, Images) of sufficient quality for clarity?

Reviewer #2: (No Response)

Reviewer #3: The analysis presented matches the analysis plan and are clearly presented

**Conclusions**

-Are the conclusions supported by the data presented?

-Are the limitations of analysis clearly described?

-Do the authors discuss how these data can be helpful to advance our understanding of the topic under study?

-Is public health relevance addressed?

Reviewer #2: (No Response)

Reviewer #3: The conclusions are supported by the data present. I have several questions on limitations:

1. Would the authors agree that the method of sampling (community workers identifying couples) have any selection biases?

2. Would including both spouses in analysis lead to some bias (assuming maybe couple-level correlations)

**Editorial and Data Presentation Modifications?**

Reviewer #2: (No Response)

Reviewer #3: I only had minor comments for this manuscript. Unfortunately I am unfamiliar with a majority of statistical approaches used in this paper, so I had provided minor comments on the study design and limitations. However, it seems like the previous reviewers have provided robust comments in these aspects.

**Summary and General Comments**

Reviewer #2: Over all what the authors replies were not satisfied me. For example. The answer to Question1 is far from satisfactory, for RPR can be positive or negative in many situations. RPR alone can’t confirm syphilis. The authors did not answer my concerns in Question 2. 

The influencing factors of fetal syphilis mainly depend on the maternal prenatal screening and proper and prompt treatment of syphilis, not RPR alone. I think the manuscript is of little significance to prevent congenital syphilis and to update the knowledge of clinicians.

Reviewer #3: (No Response)

PLOS authors have the option to publish the peer review history of their article (what does this mean?). If published, this will include your full peer review and any attached files.

Reviewer #2: No

Reviewer #3: No
---

## [Editor Report · Decision Letter 2]

12 Oct 2022

Dear Dr. Ling,

We are pleased to inform you that your manuscript 'Preconception syphilis seroprevalence and association with duration of marriage and age among married individuals in Guangdong Province, China: A population-based cross-sectional study' has been provisionally accepted for publication in PLOS Neglected Tropical Diseases.

Best regards,

Michael Marks

Section Editor

Michael Marks

Section Editor

---

## [Editor Report · Acceptance letter]

22 Nov 2022

Dear Dr. Ling,

We are delighted to inform you that your manuscript, "Preconception syphilis seroprevalence and association with duration of marriage and age among married individuals in Guangdong Province, China: A population-based cross-sectional study," has been formally accepted for publication in PLOS Neglected Tropical Diseases.

Best regards,

Shaden Kamhawi

co-Editor-in-Chief

Paul Brindley

co-Editor-in-Chief
